# FOCUS: Effective Embedding Initialization for Monolingual Specialization of Multilingual Models

**Konstantin Dobler** and **Gerard de Melo**
Hasso Plattner Institute / University of Potsdam
{konstantin.dobler, gerard.demelo}@hpi.de

## Abstract

Using model weights pretrained on a high-resource language as a warm start can reduce the need for data and compute to obtain high-quality language models for other, especially low-resource, languages. However, if we want to use a new tokenizer specialized for the target language, we cannot transfer the source model's embedding matrix. In this paper, we propose FOCUS – **F**ast **O**verlapping Token **C**ombinations **U**sing **S**parsemax, a novel embedding initialization method that initializes the embedding matrix effectively for a new tokenizer based on information in the source model's embedding matrix. FOCUS represents newly added tokens as combinations of tokens in the overlap of the source and target vocabularies. The overlapping tokens are selected based on semantic similarity in an auxiliary static token embedding space. We focus our study on using the multilingual XLM-R as a source model and empirically show that FOCUS outperforms random initialization and previous work in language modeling and on a range of downstream tasks (NLI, QA, and NER). We publish our checkpoints and code on GitHub.[1]

## 1 Introduction

Research on large language models is advancing rapidly with powerful new models being published at a break-neck pace (*e.g.,* Zeng et al., 2022a; Le Scao et al., 2022; Touvron et al., 2023). Although multilingual models have been released, many of the world's languages are not covered. Multilingual models have also been shown to have subpar performance on under-resourced languages (Wu and Dredze, 2020). Therefore, it is crucial to develop methods that harness these advances and make them available for further languages, especially low-resource ones.

A promising line of work in this regard focuses on crosslingual transfer of Transformer models pre-

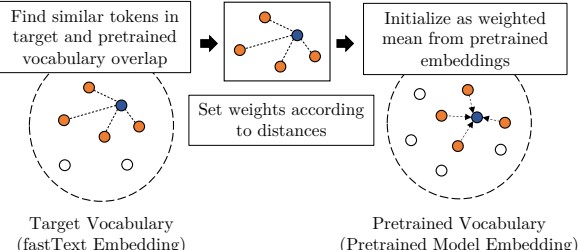

Figure 1: Illustration of FOCUS's initialization strategy for embeddings of new tokens (blue dot): Find similar tokens (orange dots) in an auxiliary fastText embedding space; then initialize the new token as their weighted mean in the pretrained embedding space.

trained on high-resource languages. Crosslingual transfer directly copies the pretrained weights in the Transformer layers to the target language model. Subsequently, the model is further adapted to the target language by continued pretraining on unlabeled target language text using the original self-supervised pretraining objective. This sort of training regimen is also known as language adaptive pretraining (LAPT; Chau et al., 2020).

However, the pretrained model's embedding matrix cannot be directly transferred if we use a new tokenizer for the target language (Artetxe et al., 2020; de Vries and Nissim, 2021). Using appropriate tokenizers has been shown to be important for the model's performance on downstream tasks (Rust et al., 2021) and is crucial if the source and target language use different scripts.

We present FOCUS, an embedding initialization method that allows us to transfer information from the source model's pretrained embedding matrix to a new embedding matrix for the target language's tokenizer. FOCUS is illustrated in Figure 1. The key idea is to use overlapping tokens between both tokenizers as anchor points and represent new target language tokens as a weighted mean of overlapping tokens' embeddings. This enables us to initialize the new embedding matrix in the same semantic space as the pretrained embedding ma-

---

[1] https://github.com/konstantinjdobler/focus

trix. We empirically show in extensive experiments across a range of different high-resource and low-resource target languages that FOCUS outperforms various strong baselines both in language modeling as well as on downstream tasks (Natural Language Inference, Question Answering, and Named Entity Recognition).

In our experiments, we focus on the multilingual XLM-R (Conneau et al., 2020) as a source model and specialize it for a single language. FOCUS is particularly well-positioned to take advantage of multilingual source models due to their larger vocabulary and the fact that they have already been pretrained to a certain extent on many potential target languages. Additionally, we show that FOCUS still improves significantly over random initialization even if only minimal vocabulary overlap is available.[2]

In previous work, a common approach has been to adapt multilingual models to target languages while simply keeping or extending the original vocabulary (Wang et al., 2019; Chau et al., 2020; Wang et al., 2020; Chau and Smith, 2021). When extending the vocabulary, FOCUS can also be applied to initialize embeddings just for the new tokens. However, we advocate considering the setting of full vocabulary replacement. Only a fraction of the multilingual vocabulary is actually used for any single language, so by fully replacing the large multilingual vocabulary with a language-specific smaller vocabulary we can enable faster training times and smaller models. XLM-R's vocabulary has 250k tokens, and replacing this with a language-specific 50k token vocabulary reduces the model size quite dramatically, by over 55%.[3] In our experiments, training with a language-specific 50k token vocabulary is 40% faster than extending the original 250k token vocabulary.[4] We summarize the contributions of our paper as follows:

- We propose FOCUS, a novel embedding initialization method that effectively transfers knowledge from a pretrained embedding matrix to one for a new, language-specific tokenizer.

- We empirically verify the effectiveness of FOCUS for language modeling and downstream

tasks in extensive experiments using XLM-R as a source model on a range of high- and low-resource languages.

- We further show that FOCUS is effective also when the target language was not part of the source models' pretraining or only a minimal vocabulary overlap is available.

## 2 FOCUS

Our goal is to initialize embeddings for tokens in a new, language-specific target vocabulary in the same semantic space as the source model's embeddings. In this study, we mainly focus on the multilingual source model XLM-R although FOCUS can in principle also be applied to monolingual source models.[5] We copy all embeddings of shared tokens between source and target tokenizer for our new embedding matrix. If the target language was already part of the source model's pretraining corpus, this takes advantage of target language tokens with pretrained embeddings in the source model's tokenizer. In any case, we take advantage of shared named entities, symbols, numbers, punctuation, and shared words resulting from code-switching between the target and pretrained vocabularies. Additional target language tokens not present in the source model are represented as a linear combination of embeddings of semantically similar shared tokens. Unlike previous work on embedding initialization, this requires neither bilingual dictionaries nor an alignment of embedding spaces across different languages (Minixhofer et al., 2022; Zeng et al., 2022b). Next, we formally describe FOCUS.

**Details of FOCUS.** We obtain as input a source vocabulary $\mathbb{V}^s$ with pretrained embeddings $\mathbf{E}^s$ and a target vocabulary $\mathbb{V}^t$ with embeddings $\mathbf{E}^t$, which we seek to initialize. The target vocabulary $\mathbb{V}^t$ is obtained by training a tokenizer on monolingual text in the target language. We use $\vec{e}_i^{\,s}$ and $\vec{e}_i^{\,t}$ to denote embeddings for individual tokens in $\mathbf{E}^s$ and $\mathbf{E}^t$, respectively. We denote the set of overlapping tokens as $\mathbb{O} = \mathbb{V}^s \cap \mathbb{V}^t$. For each overlapping token we can copy the pretrained embedding over into our target embedding matrix:

$$\forall o \in \mathbb{O}: \quad \vec{e}_o^{\,t} = \vec{e}_o^{\,s}. \tag{1}$$

[2]We study a minimal overlap consisting exclusively of tokens that are numbers, punctuation, and whitespace.

[3]Comparing 278 million parameters at $\approx$ 1.1 GB to 124 million parameters at $\approx$ 0.5 GB.

[4]Comparing a 50k token vocabulary to an extended 273k token vocabulary with HuggingFace tokenizers and transformers on two Nvidia A100 80GB GPUs.

[5]In Section 4, we also conduct experiments in a setting with almost no vocabulary overlap, showing that FOCUS can be used to transfer a model to previously unseen languages.

Note that we make an assumption here: tokens that are part of the overlap $\mathbb{O}$ have sufficiently similar semantics in our source and target vocabularies. For multilingual source models, we can exploit already existing tokens from the target language. Otherwise, this will obviously not always be the case[6], but through common named entities, code-switched tokens, symbols, numbers, and punctuation this assumption will hold reasonably often. We provide an in-depth analysis in Appendix B.

Finding an initialization in the same semantic space as the pretrained embeddings is not as easy for the set of non-overlapping ("additional") tokens $\mathbb{A} = \mathbb{V}^t \setminus \mathbb{O}$. To initialize embeddings for the additional tokens, we first train auxiliary embeddings $\mathbf{X}$ for all target tokens $\mathbb{V}^t$ (i.e., both $\mathbb{O}$ and $\mathbb{A}$).[7] In our experiments, we apply fastText on unlabeled target language data pre-tokenized with the target tokenizer for $\mathbb{V}^t$. Individual embeddings in $\mathbf{X}$ are denoted by $\vec{\boldsymbol{x}}_i$. Next, we compute the pairwise cosine similarities between the auxiliary embeddings $\vec{\boldsymbol{x}}_i$ of tokens in $\mathbb{A}$ and $\mathbb{O}$ so that for any $a \in \mathbb{A}$:

$$\vec{\boldsymbol{c}}_a = [\text{sim}(a, o_1), \ldots, \text{sim}(a, o_n)] \quad (2)$$

where $o_i$ is the overlapping token at index $i$ and:

$$\text{sim}(a, o) := \frac{\vec{\boldsymbol{x}}_a \cdot \vec{\boldsymbol{x}}_o}{\|\vec{\boldsymbol{x}}_a\|\|\vec{\boldsymbol{x}}_o\|}. \quad (3)$$

We convert the similarity scores $\vec{\boldsymbol{c}}_a$ to weights by applying sparsemax (Martins and Astudillo, 2016) over each $\vec{\boldsymbol{c}}_a$. Sparsemax is a sparse variant of softmax that assigns zero probability mass to low-probability elements, which has previously been used by Tran (2020) in a similar setting. Using sparsemax has the advantage of being able to dynamically accommodate different degrees of skew in the similarity distribution. In some cases we might have only one or two very similar tokens, in other cases, we might have significantly more. Accordingly, the weights $\vec{\boldsymbol{w}}_a$ of the overlapping tokens are:

$$\vec{\boldsymbol{w}}_a = \text{sparsemax}(\vec{\boldsymbol{c}}_a) = \underset{\vec{\boldsymbol{p}} \in \Delta}{\arg\min} \|\vec{\boldsymbol{p}} - \vec{\boldsymbol{c}}_a\|^2 \quad (4)$$

with $\Delta$ denoting the $(|\mathbb{O}|\text{-}1)$–dimensional probability simplex, i.e.,

$$\Delta := \{\vec{\boldsymbol{p}} \in \mathbb{R}^{|\mathbb{O}|} \mid \vec{\boldsymbol{1}} \cdot \vec{\boldsymbol{p}} = 1, \vec{\boldsymbol{p}} \geq \boldsymbol{0}\}. \quad (5)$$

We then initialize the target embeddings for each additional token $a$ as a weighted mean over pretrained embeddings of the overlapping tokens from $\mathbf{E}^s$, with the weights given by $\vec{\boldsymbol{w}}_a$. Due to sparsemax, most of the elements in each $\vec{\boldsymbol{w}}_a$ will be zero. Note that we use the pretrained embeddings $\mathbf{E}^s$ instead of the auxiliary embeddings $\mathbf{X}$, as only the pretrained embeddings are in the same semantic space as the rest of the transferred Transformer layers. Therefore:

$$\forall a \in \mathbb{A}: \quad \vec{\boldsymbol{e}}_a^t = \sum_{o \in \mathbb{O}} w_{a,o} \, \vec{\boldsymbol{e}}_o^s. \quad (6)$$

**Summary.** FOCUS uses cheap and fast-to-train static embeddings for tokens in the target vocabulary to select semantically similar overlapping tokens for each additional target token. The pretrained embeddings of the overlapping tokens are then used to initialize embeddings for the additional target tokens. In Appendix B, we provide further implementation details as well as a detailed analysis of the different types of overlapping tokens we encountered in our experiments.

## 3 Experimental Setup

We perform experiments using XLM-R as our multilingual source model, due to its popularity and widespread use.[8] We use the base variant for all experiments. Our language-specific tokenizers are trained in the same way as XLM-R for comparability, specifically SentencePiece tokenization (Kudo and Richardson, 2018) with the Unigram algorithm (Kudo, 2018). We use HuggingFace tokenizers and a vocabulary size of 50k tokens for all languages.

### 3.1 Baselines

To evaluate FOCUS, we compare against multiple strong baselines for embedding initialization as well as other methods of adapting XLM-R to a target language. We always transfer all layers of XLM-R, except for the embedding. Minixhofer et al. (2022) already demonstrate the superiority of this over random initialization of all weights, so we do not compare against the weak baseline of training a model completely from scratch.

**XLM-R with the original vocabulary.** We report results of using XLM-R off-the-shelf without

---

[6] Consider words known as *false friends*: words with the same spelling but different meanings across languages.

[7] In principle, we could instead also obtain token embeddings from already pretrained fastText word embeddings following Minixhofer et al. (2022). We show in Section 4 that training fastText directly at the token level provides a better initialization.

[8] As of October 2023, XLM-R has 12.2 million downloads per month on the HuggingFace Hub.

language-adaptive pretraining (LAPT) as well as after adapting XLM-R to the target language with the original vocabulary kept as-is.

**Random Initialization.** For vocabulary replacement with a language-specific tokenizer and random embedding initialization, we copy the original pretrained embeddings following Zoph et al. (2016). This randomly maps pretrained embeddings to tokens in the new vocabulary and performed slightly better than other types of random initialization in preliminary experiments.[9] In this case, we also consider the variant of training just the embeddings for an additional 20% of training steps before unfreezing the rest of the network (called 2-STAGE-LAPT). De Vries and Nissim (2021) note that this allows the new embeddings to adapt to the transferred Transformer layers to prevent catastrophic forgetting. Therefore, this strong baseline is trained 20% longer than other methods.

**WECHSEL.** We additionally compare against using WECHSEL (Minixhofer et al., 2022) to initialize the embedding matrix for the language-specific tokenizer. WECHSEL is a method for embedding initialization originally designed for transferring monolingual source models. It relies on aligning pretrained word embeddings for the source and target languages using the Orthogonal Procrustes method (Schönemann, 1966) with bilingual dictionaries as seed data. Then, each source and target token is embedded into the same semantic space using the out-of-vocabulary method of fastText, resulting in aligned static token embeddings for both languages.

To faithfully apply WECHSEL with a multilingual source model, we would need to provide a word embedding space for all the languages that are part of the multilingual models' pretraining corpus. Also, gathering bilingual dictionaries from all source languages to the target language would become a challenge. Instead, we apply WECHSEL as-is using only pretrained English fastText word embeddings for the source model. This effectively assumes that all pretrained source token embeddings are English, which is a rough but not entirely unreasonable assumption given the predominance of English over other languages in the pretraining corpus of XLM-R. We can further commit to this assumption by deleting all non-English tokens from

| Language | Dataset Size (GB) |
|----------|-------------------|
| German   | 18 GB             |
| Arabic   | 5.4 GB            |
| Kiswahili | 0.3 GB           |
| Hausa    | 0.06 GB           |
| isiXhosa | 0.03 GB           |

Table 1: Size of datasets from CC100 used for LAPT.

the pretrained vocabulary before applying WECHSEL, which we dub WECHSEL$_{EN}$. This yields an initialization method similar to the mixture mapping method proposed by Wang et al. (2019).

**Vocabulary Extension.** We also run experiments with vocabulary extension following Wang et al. (2020) by extending with the top 30k tokens of the language-specific tokenizer as well as using FOCUS to initialize embeddings for the extended tokens.

### 3.2 Language-Adaptive Pretraining (LAPT)

For LAPT, we use the same self-supervised Masked Language Modeling (MLM) objective as in the original pretraining of XLM-R. We use the CC100 corpus to obtain unlabeled text in our target languages, which was also already used for the pretraining of XLM-R (Conneau et al., 2020). Therefore, we do not introduce any new unseen data. We show dataset sizes for our target languages in Table 1. We use the same hyperparameters for all languages, as detailed in Appendix A. In particular, we use a batch size of 128 with chunked sequences of 256 tokens and train our models on 50 million samples (resulting in a total of 12.8 billion training tokens and 390k optimizer steps).

### 3.3 Evaluation

We also evaluate our models on downstream tasks in their respective target languages. We perform downstream task evaluation on five languages: German, Arabic, Kiswahili, isiXhosa, and Hausa. They were chosen to provide a mix of high-, medium- and low-resource languages, typological and script diversity while satisfying the practical constraints of available evaluation datasets. We refer to German as high-resource, Arabic and Kiswahili as medium-resource, and isiXhosa and Hausa as low-resource languages.

We use the translated training sets of XNLI (Conneau et al., 2018) to evaluate Natural Language Inference (NLI) in the translate–train setting. To

---

[9]We refer to this as random initialization. Random mapping performed slightly better than initialization from a normal or uniform distribution.

evaluate Question Answering, we use German-QuAD (for German, Möller et al., 2021) and Ty-DiQA GoldP (for Swahili and Arabic; Clark et al., 2020). We perform Named Entity Recognition (NER) experiments using the balanced `train–dev–test` split of WikiANN (Rahimi et al., 2019; Pan et al., 2017). Additionally, we evaluate NER for German on the GermEval2014 dataset (Benikova et al., 2014) and for Swahili, Hausa, and isiXhosa using MasakhaNERv2 (Adelani et al., 2022). If there is no dedicated dev split, we construct our own with a random sample of 10% of the training data. We perform model selection on the dev split and report the selected checkpoint's result on the test set. We report accuracy for XNLI and $F_1$-scores otherwise. We run all experiments five times with different random seeds and report the mean and standard deviation. Hyperparameters for all evaluation tasks are given in Appendix A.

Furthermore, we evaluate the initialization performance of FOCUS without further training measured by the MLM loss on a held out set on five additional very low-resource languages (Scottish Gaelic, Luxembourgish, Cebuano, Samoan, and Hmong). For these languages, we use mC4 (Raffel et al., 2020) and OSCARv23.01 (Abadji et al., 2022) as additional data sources for unlabeled text.

## 4 Results

We present downstream task results for NLI and QA in Table 2 and for NER in Table 3. In the following, we discuss various aspects of these results. In Figure 2, we show loss curves on a held out set when adapting XLM-R with custom tokenizers. In Table 4, we report the masked language modeling (MLM) loss of various methods right after initialization (no further training performed).

**Effectiveness of FOCUS.** Table 4 shows the effectiveness of FOCUS initialization for vocabulary replacement. Directly after initialization without further training, FOCUS significantly outperforms all other initialization methods. In Figure 2, we show loss curves over the course of language-adaptive pretraining (LAPT). For random initialization, we only plot the second stage after the embeddings have already been trained for an additional 20% of total training steps. FOCUS yields a lower loss than random initialization even at the end of training, despite random initialization having been trained for more steps in total. WECHSEL starts off worse than FOCUS but catches up over the course of train-ing. Naturally, the effect of initialization is less pronounced the longer we train the models. We have deliberately constructed a difficult evaluation with our long training regime of 12.8 billion tokens. In settings where less compute is available, FOCUS may be even more beneficial.

The improved effectiveness on the pretraining objective also translates to gains in downstream tasks, as reported in Table 2 and Table 3. FO-CUS initialization outperforms random initialization across all downstream tasks and languages (except for Arabic TyDiQA). WECHSEL also improves over random initialization, but FOCUS obtains superior results. FOCUS can also be applied for vocabulary extension instead of vocabulary replacement. Here, we see less of an improvement over the random initialization baseline. This could be due to the smaller impact of FOCUS, since only a relatively small percentage of the large extended vocabulary is affected.

**Vocabulary Extension or Replacement?** We find that vocabulary extension generally performs worse on downstream tasks than keeping the original vocabulary. This finding is in line with results reported by Ebrahimi and Kann (2021) on a set of 30 typologically diverse languages. Prior studies proposing vocabulary extension (Chau et al., 2020; Wang et al., 2020; Chau and Smith, 2021) used mBERT and were motivated by the possibility of out-of-vocabulary (OOV) tokens. For XLM-R using SentencePiece with 100% character set coverage or byte-level tokenizers, OOV tokens can always be represented at the character or byte level. Therefore, the benefits of vocabulary extension might be less pronounced in these cases because the OOV problem is less relevant to begin with.

On average, when combined with FOCUS initialization, vocabulary replacement outperforms both vocabulary extension and keeping the original vocabulary. Nevertheless, keeping the original vocabulary intact proves to be a strong baseline and for the high-resource language German even outperforms vocabulary replacement with FOCUS. However, vocabulary replacement paired with FO-CUS performs better on medium- and low-resource languages, results in smaller models, and is thus faster to train.

**Low-Resource Languages.** Focusing on lesser-resourced languages, FOCUS outperforms random initialization and LAPT with the original vocabu-

| Method | XNLI (translate-train) | | | | GermanQuAD / TyDiQA | | | |
|---|---|---|---|---|---|---|---|---|
| | German | Arabic | Kiswahili | Avg. | German | Arabic | Kiswahili | Avg. |
| XLM-R (original vocab) | | | | | | | | |
| - off-the-shelf | 78.8 ± 0.3 | 74.8 ± 0.4 | 69.1 ± 0.4 | 74.2 | **_71.3_** ± 0.4 | 78.4 ± 0.9 | 73.9 ± 1.4 | 74.5 |
| - LAPT | **_78.9_** ± 0.4 | **75.1** ± 0.6 | **72.4** ± 0.4 | **75.5** | 70.5 ± 0.8 | **78.9** ± 0.5 | **75.8** ± 1.0 | **75.0** |
| XLM-R (replaced vocab) | | | | | | | | |
| - Random + LAPT[†] | 77.6 ± 0.4 | 74.6 ± 0.4 | 71.2 ± 0.3 | 74.5 | 69.1 ± 0.7 | 79.3 ± 0.6 | 74.2 ± 1.0 | 74.2 |
| - WECHSEL$_{EN}$ + LAPT | 77.7 ± 0.5 | 75.4 ± 0.3 | 72.0 ± 0.2 | 75.0 | 71.0 ± 0.4 | 79.3 ± 1.0 | 75.2 ± 0.7 | 75.2 |
| - WECHSEL + LAPT | 78.2 ± 0.2 | 76.0 ± 0.2 | 72.3 ± 0.3 | 75.5 | 70.5 ± 0.5 | _79.4_ ± 0.9 | 75.5 ± 1.5 | 75.1 |
| - FOCUS + LAPT | **78.3** ± 0.6 | **_76.5_** ± 0.4 | **_72.9_** ± 0.5 | _75.9_ | **71.3** ± 0.2 | 79.1 ± 0.4 | **76.5** ± 1.5 | _75.6_ |
| XLM-R (extended vocab) | | | | | | | | |
| - Random + LAPT | 77.7 ± 0.6 | 75.2 ± 0.6 | 71.8 ± 0.4 | 74.9 | **69.8** ± 0.6 | 77.7 ± 0.6 | 76.3 ± 0.8 | 74.6 |
| - FOCUS + LAPT | **78.0** ± 0.4 | **75.5** ± 0.4 | **72.1** ± 0.2 | **75.2** | 69.5 ± 0.3 | **77.8** ± 1.0 | **_77.0_** ± 0.6 | **74.7** |

Table 2: Results on Natural Language Inference and Question Answering tasks. Details on the datasets used for evaluation are given in Section 3.3. We **bold** the best result in each section and underline the overall best result. LAPT is short for language-adaptive pretraining; we perform LAPT for 50 million samples on unlabeled target texts. [†]: For random initialization, we train just the embeddings for an additional 20% of training steps before full LAPT to create a stronger baseline.

| Method | WikiANN | | | | GermEval14 / MasakhaNERv2 | | | | |
|---|---|---|---|---|---|---|---|---|---|
| | German | Arabic | Kiswahili | Avg. | German | Kiswahili | Hausa | isiXhosa | Avg. |
| - off-the-shelf | 86.3 ± 0.2 | 85.7 ± 0.3 | 86.6 ± 0.5 | 86.2 | 85.6 ± 0.3 | 92.0 ± 0.1 | 84.2 ± 0.5 | 85.5 ± 0.3 | 87.1 |
| - LAPT | **86.7** ± 0.1 | 87.1 ± 0.1 | 86.9 ± 0.6 | 86.9 | **86.8** ± 0.2 | 92.5 ± 0.2 | 85.6 ± 0.4 | 88.3 ± 0.2 | 88.3 |
| XLM-R (replaced vocab) | | | | | | | | | |
| - Random + LAPT[†] | 86.0 ± 0.1 | 87.5 ± 0.1 | 85.8 ± 0.5 | 86.4 | 85.9 ± 0.3 | 92.3 ± 0.2 | 85.0 ± 0.3 | 87.4 ± 0.2 | 87.8 |
| - WECHSEL$_{EN}$ + LAPT | 86.4 ± 0.1 | 87.8 ± 0.1 | 86.6 ± 0.9 | 87.0 | 86.4 ± 0.2 | 92.3 ± 0.1 | –[‡] | –[‡] | – |
| - WECHSEL + LAPT | 86.5 ± 0.2 | _87.9_ ± 0.3 | _87.4_ ± 0.6 | _87.3_ | **86.7** ± 0.1 | 92.2 ± 0.1 | –[‡] | –[‡] | – |
| - FOCUS + LAPT | **86.6** ± 0.2 | _87.9_ ± 0.1 | 86.9 ± 0.4 | 87.1 | 86.6 ± 0.0 | _92.6_ ± 0.1 | **_86.0_** ± 0.4 | **_88.5_** ± 0.4 | _88.4_ |
| XLM-R (extended vocab) | | | | | | | | | |
| - Random + LAPT | 85.6 ± 0.2 | 85.2 ± 0.3 | **86.2** ± 0.7 | 85.6 | 85.4 ± 0.3 | 92.0 ± 0.2 | 84.1 ± 0.2 | 87.2 ± 0.4 | 87.5 |
| - FOCUS + LAPT | **86.0** ± 0.1 | **85.3** ± 0.3 | 86.2 ± 0.3 | **85.8** | **85.6** ± 0.2 | **92.1** ± 0.2 | **84.9** ± 0.4 | **87.7** ± 0.3 | **87.9** |

Table 3: Results on Named Entity Recognition (NER) tasks. Details on the datasets used for evaluation are given in Section 3.3. We **bold** the best result in each section and underline the overall best result. [†]: For random initialization, we train just the embeddings for an additional 20% of training steps before full LAPT to create a stronger baseline. –[‡]: Languages not covered by the pretrained fastText word embeddings used by WECHSEL.

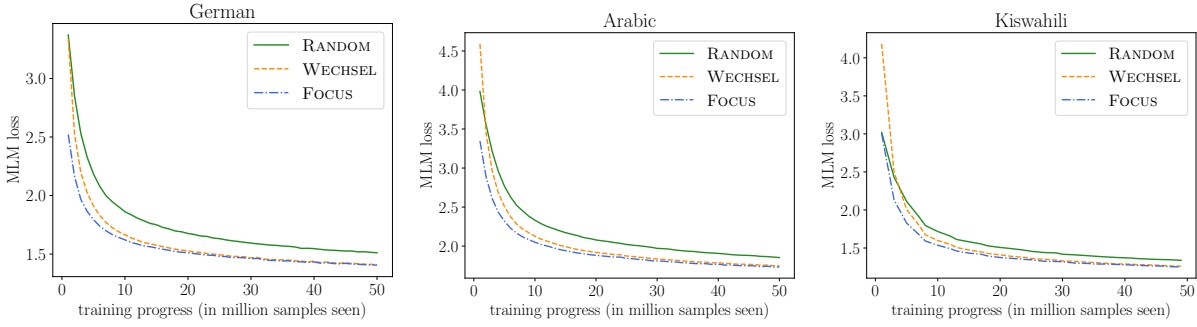

Figure 2: Masked Language Modeling (MLM) loss of different methods for vocabulary replacement over the course of further pretraining (LAPT), evaluated on a held out set. The first data point is logged at 1 million samples. For random initialization, we plot only the second stage, i.e., after already training just the embeddings for 10 million samples. This allows us to compare FOCUS and WECHSEL embedding initialization directly with gradient descent training of the embeddings.

| | Part of XLM-R's pretraining | | | | | | Not Part of XLM-R's pretraining | | | |
| Data source → | CC100 | | | | | | OSCARv23.01 | | mC4 | |
| Initialization ↓ | German | Arabic | Kiswahili | isiXhosa | Hausa | Scott. Gaelic | Luxembourgish | Cebuano | Samoan | Hmong |
|---|---|---|---|---|---|---|---|---|---|---|
| Random | 24.0 | 24.1 | 24.2 | 25.5 | 23.7 | 22.8 | 24.4 | 22.5 | 21.9 | 22.8 |
| Focus | **4.0** | **5.2** | **4.8** | **7.6** | **6.0** | **6.1** | **8.2** | **6.3** | **4.9** | **5.7** |
| - Word-fastText | 4.3 | 5.5 | 6.0 | –[†] | –[†] | 7.5 | 8.7 | 6.7 | –[†] | –[†] |
| - Symbolic Overlap | 10.6 | 10.6 | 10.7 | 10.4 | 10.4 | 10.4 | 10.2 | 9.7 | 8.4 | 8.1 |
| Wechsel | 8.3 | 9.8 | 11.2 | –[†] | –[†] | 11.1 | 10.9 | 10.1 | –[†] | –[†] |

Table 4: MLM loss on a held-out set immediately after initialization (no training performed) with full vocabulary replacement. We use the same vocabulary for all methods in a single language. Symbolic Overlap restricts overlapping tokens to numbers, punctuation, or whitespace. Word-fastText uses Wechsel's method of turning pretrained word embeddings into token embeddings instead of our proposed directly trained token-level embeddings. –[†]: Languages are not covered by the pretrained fastText word embeddings used by Wechsel.

lary on NER for Hausa and isiXhosa. Furthermore, we report the Masked Language Modeling loss directly after initialization on a number of very low-resource languages in Table 4. We see that across all languages, including the very low-resource ones, Focus achieves the best results. Focus also provides a good initialization even when the target language was not part of the source model's pretraining.

In low-resource settings, a key advantage of Focus is that we only need unlabeled target language data to train our auxiliary embeddings – a resource already needed for Lapt in any case. Unlike Wechsel, no bilingual dictionary is required, the quality and coverage of which might also be insufficient in low-resource settings. Some low-resource languages, such as Hausa and isiXhosa, are also not covered by Wechsel's source of pretrained word embeddings.[10]

**Effect of Vocabulary Overlap.** Naturally, the quality and quantity of overlapping tokens influences the success of Focus. To investigate this, we conducted empirical analyses in two settings: using the full overlap and using only overlapping tokens that are symbols, numbers, or punctuation (Symbolic Overlap). Full overlap can take advantage of the source model's multilingual pretraining if the target language or a closely related language were part of the pretraining corpus. In any case, however, symbolic tokens such as whitespace, numbers, and punctuation should generally be available, allowing us to transfer a model to any language. In Table 4, we show that even when using only symbolic overlapping tokens, Focus outperforms Wechsel on medium to low-resource languages (e.g., Scottish Gaelic, Luxembourgish, Kiswahili,

and others). For Arabic and German, Focus with only symbolic overlapping tokens performs slightly worse than Wechsel. In practice however, we will generally have numerous further overlapping tokens such as named entities and code-switched tokens. This is demonstrated by our results for Luxembourgish, Cebuano, Samoan, and Hmong – all languages that XLM-R and XLM-R's tokenizer were not pretrained on. Here, using the full overlap outperforms using only symbols, suggesting more beneficial overlapping tokens beyond the ones included in our symbolic overlap. Overall, these results show that Focus can provide a good initialization even when the target language was not part of the source model's pretraining.

**Auxiliary Embeddings.** Wechsel proposes a method to use pretrained word-level fastText embeddings to obtain token-level embeddings. We propose to directly train token-level fastText embeddings. In Table 4, we additionally show Focus's initialization performance when using the Wechsel-style method to obtain token-level fastText embeddings (Word-fastText). We see that using our directly trained token-level fastText embeddings results in a better initialization for low- and high-resource languages.

**Wechsel_EN.** On average, Wechsel actually fares slightly better than Wechsel_EN, although Wechsel_EN also improves over random initialization. For Wechsel_EN, we followed Wang et al. (2019) in selecting English tokens in XLM-R's original vocabulary by taking the overlap with a language-specific English tokenizer's vocabulary. Due to the substantial presence of English in XLM-R's original vocabulary, this may have been too restrictive, excluding too many potentially useful tokens.

---

[10] https://fasttext.cc/docs/en/crawl-vectors.html

# 5   Related Work

We now discuss further related work apart from the studies introduced in Section 1.

**Language Adaptive Pretraining (LAPT).** Alabi et al. (2022) adapted XLM-R to up to 20 African languages at the same time instead of specializing on a single language. Ebrahimi and Kann (2021) and Wang et al. (2022) used resources with much higher language coverage than web-scraped monolingual texts (the Bible and lexicons, respectively) to adapt pretrained multilingual models to unseen languages. Muller et al. (2021) transliterated unseen languages into Latin script to improve the results when using an existing pretrained vocabulary.

**Adapters.** In contrast to approaches changing all pretrained model weights, Pfeiffer et al. (2020) introduce additional adapter modules and only these new weights are changed. This is more parameter-efficient than full model adaptation, but gradients still need to be backpropagated throughout the model until the first adapter (Rücklé et al., 2021). Also, adapters introduce additional computational cost at inference time.

**Bilingual Embedding Alignment.** Vernikos and Popescu-Belis (2021) propose SMALA to calculate a mapping between embedding spaces for two languages to find semantically similar tokens across languages. They also experiment with initializing the remaining tokens based on this cross-lingual mapping. WECHSEL (Minixhofer et al., 2022) aligns word embeddings from two different languages. Such alignments operate under the assumption of near-isomorphism between embedding spaces of different languages (Vulić et al., 2020), i.e., that they share a similar geometric structure. Recent studies have challenged this assumption, especially for language pairs with typological (Søgaard et al., 2018; Patra et al., 2019; Ormazabal et al., 2019) and resource (Vulić et al., 2020; Fu et al., 2020) differences. This is especially detrimental in the case of language model transfer, as we usually transfer from a high-resource language such as English to less-resourced languages with potentially different typology. FOCUS does not require the alignment of embedding spaces.

For multilingual source models, WECHSEL also disregards a valuable resource at our disposal: target language tokens that already have pretrained embeddings in the multilingual source model. For these tokens, we can copy their pretrained embeddings as a gold standard. Obtaining a different initialization is likely to lead to a worse result. FOCUS is well-positioned to take advantage of these pretrained embeddings of target language tokens.

Additionally, WECHSEL requires a bilingual dictionary as an additional resource to seed the embedding space alignment. For low-resource languages, such a bilingual dictionary might be of lower quality or not available. FOCUS does not require bilingual dictionaries as an additional resource.

**Other Embedding Initialization Methods.** In concurrent work, Ostendorff and Rehm (2023) propose a similar method to FOCUS that initializes an embedding matrix for a new vocabulary based on combinations of overlapping tokens with a pretrained embedding matrix, but use the embedding layer of a smaller pretrained Transformer model instead of static fastText embeddings as an auxiliary embedding space. However, their study only provides results on the high-resource language German as a target language and they do not consider BERT-style source models. If no smaller pretrained Transformer model with the desired tokenizer is available, training one from scratch comes with a much higher computational cost than training the fastText embeddings for FOCUS. Zeng et al. (2022b) create a new vocabulary and embedding matrix for the target language by translating tokens in the source vocabulary with bilingual dictionaries.

# 6   Conclusion

We propose FOCUS, a novel embedding initialization method for the monolingual specialization of language models with a language-specific tokenizer. FOCUS uses the vocabulary overlap between source and target languages to effectively transfer the pretrained embeddings to the new target tokenizer's embedding matrix. In a series of experiments across a diverse set of languages and several different tasks, we show that FOCUS outperforms other available embedding initialization methods without requiring additional resources like bilingual dictionaries. FOCUS can provide a good initialization even if only a minimal vocabulary overlap is available and when the target language has not been part of the source model's pretraining. We release our code and model checkpoints on GitHub.[11]

---

[11] https://github.com/konstantinjdobler/focus

## Limitations

We evaluate FOCUS only for BERT-like Transformer models. In principle, the method is applicable to any model that uses a vocabulary, tokenizer, and embedding matrix. In future work, we hope to investigate the use of FOCUS on GPT decoder models, as explored by Ostendorff and Rehm (2023).

We conduct downstream task evaluations for NLI, QA, and NER on German, Arabic, and Swahili. For the low-resource languages isiXhosa and Hausa, we conduct downstream task experiments for NER. This provides a good mix of different levels of available resources, scripts, and typology. However, further evaluations on languages covering more scripts and languages that were not part of the source models' pretraining are needed to substantiate the effectiveness of FOCUS in these settings. All our chosen languages have monolingual texts available for further pretraining. As Wang et al. (2022) note, this is not the case for many other low-resource languages. Since further pretraining on target language data is a key component of our model adaptation strategy, the applicability of FOCUS is also limited in this regard, although such data can in some cases also be synthesized.

## Ethics Statement

In this work, we conduct the main part of our downstream task experiments on German, Arabic, and Swahili. These choices stem from our desire to provide practically useful ideas that reflect the current availability of models and to conduct experiments on downstream tasks such as question answering, NLI, and named entity recognition, for which we need relevant ground truth data.

Finally, researchers and practitioners need to be cognizant of the fact that adopting existing monolingual or even multilingual models as a starting point instead of training new models from scratch can lead to remnant biases towards the original pretraining data. Hence, there is a risk that a model adopts certain forms of behavior that reflect other languages and cultures than that of the language community one is targeting. Also, web-scale datasets used for pretraining such as CC100 might contain personal and sensitive information. Such behavior needs to be assessed very carefully before any real-world deployment of the models.

## Acknowledgements

The authors acknowledge the financial support by the German Federal Ministry for Education and Research (BMBF) through the project «KI-Servicezentrum Berlin Brandenburg» (01IS22092). We also thank the reviewers for their helpful comments.

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

## A  Hyperparameters and Experiment Details

We conducted all experiments on a heterogeneous compute cluster with Nvidia V100 32GB, A100 40GB, A100 80GB, and A6000 48GB GPUs. Depending on availability we used one, two, or four GPUs for our experiments and adjusted the batch size per device so that we retain the same effective batch size. Depending on total model size, we also used gradient accumulation with smaller batch sizes to fit the model into GPU memory. We used PyTorch (Paszke et al., 2019) and pytorch-lightning (Falcon et al., 2019) as well as the HuggingFace transformers (Wolf et al., 2020), tokenizers (HuggingFace, 2021) and datasets (Lhoest et al., 2021) libraries. We used the fp16 mixed precision training implemented by pytorch-lightning.

**Further pretraining.**  We used the same hyperparameters for all target languages, as detailed in Table 5. We trained for a total of 50 million samples with batches of 128 sequences of 256 tokens. This results in a total of 390,625 optimizer steps (weight updates). We used AdamW optimization (Loshchilov and Hutter, 2019) as implemented in torch.optim[12] and a linear learning rate warmup for 5 million samples (39,062 optimizer steps) followed by a constant learning rate at $5 \times 10^{-5}$. We used a constant schedule to allow for more flexible experimentation regarding the total number of training steps and to ensure that the impression of converged loss curves is not a false positive induced by a decaying learning rate. We also conducted preliminary experiments using a cosine learning rate schedule and did not observe a significant difference. We used the CC100 dataset in line with its intended use for the pretraining of language models. For German, our training does not even complete a full epoch.

**Datasets for low-resource languages.**  Comparing OSCARv23.01 and mC4 as a data source for low-resource languages, we observe that for Corsican[13], Cebuano and Luxembourgish, the quality in mC4 is quite poor. Training a tokenizer on these datasets results in a tokenizer that, on average, encodes *fewer* characters per token than when

---

[12]Since we do not use weight decay, this is equivalent to using Adam (Kingma and Ba, 2015).

[13]Results not included in the paper as the language is not provided by OSCARv23.01.

| Hyper-parameter | Value |
|---|---|
| peak learning rate | $5 \times 10^{-5}$ |
| learning rate schedule | constant |
| learning rate warmup | 5 million samples |
| batch size | 128 |
| sequence length | 256 |
| gradient clipping | 1.0 |
| Adam $\epsilon$ | $1 \times 10^{-8}$ |
| Adam $\beta_1$ | 0.9 |
| Adam $\beta_2$ | 0.999 |
| training samples | 50 million |
| resulting train steps | 390625 |

Table 5: Hyper-parameters for further pretraining on target language data.

using the original XLM-R tokenizer. Motivated by this, we turned to OSCARv23.01 (for Cebunao and Luxembourgish, since it does not contain Corsican). We filter each dataset based on the quality warnings header, footer, and noisy provided by OSCARv23.01. Training a tokenizer on this data yielded the expected results. For Hmong and Samoan, we did not observe such degraded tokenization training on mC4. Nevertheless, these corpora (and all web-crawled corpora of low-resource languages) can also be expected to be noisy.

**Downstream tasks.**  We detail our hyperparameters for all downstream tasks in Table 6. We largely followed default values of finetuning scripts provided by Huggingface[14], but adjusted the training epochs depending on dataset size, added a linear learning rate warmup for 10% of total training steps, and adjusted the batch size based on used GPU memory per task. Additionally, we used a $2 \times 10^{-5}$ peak learning rate for all non-QA tasks. We repeated each experiment five times with the random seeds {1,2,3,4,5} and report the mean and standard deviation across runs. For XNLI, we report accuracy, for TyDiQA, GermanQuAD, WikiANN, MasakhaNERv2, and GermEval14, we report the $F_1$-Score.

**Tokenizer training.**  Our language-specific tokenizers were trained in the same way as XLM-R for comparability, specifically SentencePiece tokenization (Kudo and Richardson, 2018) with the Unigram algorithm (Kudo, 2018). We used Hug-

---

[14]github.com/huggingface/transformers/tree/main/examples/pytorch

gingFace `tokenizers` and a vocabulary size of 50k tokens for all languages. The resulting vocabularies contain a large amount (roughly 10k tokens) of emojis and Chinese, Japanese, and Korean single characters, which is an artifact of SentencePiece's `character_coverage` parameter (which defaults to 100%). Characters are included in the vocabulary even if they appear only once in the large amount of noisy web-scraped training documents. This effectively means that our language-specific vocabularies are roughly 10k tokens smaller in practice, as such single characters rarely occur in the training data. In practice, one may wish to tune the `character_coverage` carefully based on the requirements of the target language if a smaller model is desired.

## B Further details for FOCUS

**FastText training.** To obtain static token embeddings for FOCUS, we train fastText embeddings on tokenized target language training data. We mostly used default hyper-parameters but increased the dimensionality to 300, as is commonly done in the literature (Mikolov et al., 2013; Bojanowski et al., 2017). We ran the training for three epochs. On German, due to its corpus size, we ran only a single epoch.

Additionally, we set a `minCount` of 10 for tokens during fastText training to filter out very rare tokens. These rare tokens are initialized from a normal distribution with mean and standard deviation per dimension calculated from the source embedding matrix, as done by WECHSEL for tokens that have no subwords in the pretrained word embedding. Setting `minCount` also helps with filtering the noisy single characters that are part of our tokenizers due to SentencePiece's `character_coverage` parameter.

**Vocabulary Overlap.** FOCUS relies on overlapping tokens between the new and pretrained vocabularies. Ideally, an overlapping token would have the same semantics in the target language vocabulary and in the pretrained vocabulary. If the target language was already part of the pretraining, this is most obviously true for (sub-)words that only occur in the target language. Differences in script or peculiarities of the target language (such as German umlauts and other language-specific accented characters) help facilitate such occurrences. In many languages, especially online, there is widespread code-switching with English, leading to English words being interspersed within native sentences, which also contributes to shared semantics. A considerable share of tokens is also made up of names, named entities, symbols, numbers, and punctuation. While these are not exclusive to any particular language, they are likely to possess the same semantics across languages, making them good overlapping tokens. We report the number of overlapping tokens for languages used during training in Table 8.

Additionally, we manually classified a random sample of 500 overlapping tokens for German and report the results in Table 7. The overlap is calculated between XLM-R's original tokenizer and our newly trained, language-specific German one. For this analysis, we excluded the noisy single-character tokens mentioned in Appendix A. We conclude that a considerable share of the overlapping tokens for German does indeed possess similar semantics in the pretrained and new vocabularies. For less-resourced languages than German that were still part of the multilingual models' pretraining, we can expect fewer overlapping tokens that are directly part of the target language. High-resource languages have a larger share of language-specific tokens in the vocabulary of XLM-R. However, for languages with an uncommon or unique script, tokens are more likely to be exclusive to the target language. During the pretraining of XLM-R, low-resource languages are also oversampled (Conneau et al., 2020). Therefore, tokens that are shared between low and high-resource languages are more likely to also have the low-resource language semantics encoded in their embeddings than would otherwise be the case.

**Overlaps between different tokenizers.** In general, we only consider tokens as overlapping if they are an exact match (including case and the "beginning of word" (BOW) signifier. However, for tokens that only consist of numbers, punctuation, or whitespace, we implement fuzzy matching where we disregard the case and the BOW signifier.

A peculiarity of calculating token overlaps between different *kinds* of tokenizers is the representation of tokens that are BOW tokens and non-ASCII characters. For example, the HuggingFace implementation of Byte-Level BPE uses Ġ as a prefix for BOW tokens, whereas XLM-R's tokenizer uses _. To complicate things, BERT's tokenizer WordPiece (Devlin et al., 2019) prefixes tokens that are *not* BOW with ##. Also, Byte-level BPE represents non-ASCII characters in tokens differ-

| Hyper-parameter | XNLI | QA | WikiANN | GermEval14 | MasakhaNERv2 |
|---|---|---|---|---|---|
| epochs | 2 | 5* | 5* | 25 | 25 |
| peak learning rate | $2 \times 10^{-5}$ | $5 \times 10^{-5}$ | $5 \times 10^{-5}$ | $5 \times 10^{-5}$ | $2 \times 10^{-5}$ |
| lr schedule | linear | linear | linear | linear | linear |
| lr warmup ratio | 10% | 10% | 10% | 10% | 10% |
| batch size | 128 | 64 | 128 | 128 | 128 |
| sequence length | 256 | 384 | 256 | 256 | 256 |
| gradient clipping | 1.0 | 1.0 | 1.0 | 1.0 | 1.0 |
| Adam $\epsilon$ | $1 \times 10^{-8}$ | $1 \times 10^{-8}$ | $1 \times 10^{-8}$ | $1 \times 10^{-8}$ | $1 \times 10^{-8}$ |
| Adam $\beta_1$ | 0.9 | 0.9 | 0.9 | 0.9 | 0.9 |
| Adam $\beta_2$ | 0.999 | 0.999 | 0.999 | 0.999 | 0.999 |

Table 6: Hyper-parameters for our downstream tasks. *: After observing high variance due to smaller training set sizes, we adjusted the number of epochs for Swahili to 25 for QA (on TyDiQA GoldP) and NER (on WikiANN). We evaluate every 5% of total training steps and report the best checkpoint's results on the test set.

| Category | Share | Examples |
|---|---|---|
| Symbols & Numbers | 9% | 3.5, 1919, 3500, ..., ;-), !!!, 1%, [6] |
| Names & Entities | 10% | BlackBerry, Oscar, Messi, JavaScript |
| German (sub-)words | 46% | Bewerber, Wahl, günstig, fallen |
| English & Code-switched | 18% | Smoothie, FAQ, Backup, Settings |
| Not assignable | 17% | ik, Kri, kub, rez, zy, BF, oka |

Table 7: Investigation of overlapping tokens on German. The evaluation was conducted manually by one of the authors on a random sample of 500 overlapping tokens. In the examples, leading spaces are omitted and we further exclude the "noisy" single-character tokens mentioned in Appendix A.

|  | Tokens in Overlap | |
|---|---|---|
| Language | minCount = 10 | Full |
| German | 14,485 | 18,986 |
| Arabic | 10,658 | 13,996 |
| Swahili | 10,443 | 12,353 |
| Hausa | 11,481 | 14,806 |
| isiXhosa | 6,222 | 8,333 |

Table 8: Number of tokens in the overlap between language-specific and XLM-R's original vocabulary. For learning fasttext embeddings with FOCUS, we set minCount = 10, which filters out very rare and noisy tokens.

ently than XLM-R's tokenizer. In our experiments in this paper, we only use the XLM-R tokenizer, which also matches the source model's tokenizer, and therefore avoid these problems. However, a correct canonicalization of tokens to a common form is crucial to enable FOCUS when the tokenizers of source and target model do not match. We implement such a canonicalization method for common tokenizers and release it as part of our ready-to-use implementation of FOCUS.[15]

[15]https://github.com/konstantinjdobler/focus