# OpenReview forum: "FOCUS: Effective Embedding Initialization for Monolingual Specialization of Multilingual Models"
_EMNLP/2023/Conference — EMNLP 2023 Main_

### Official Review · Reviewer_s1L6 · 2023-08-03

**Soundness:** 4

**Excitement:**

4: Strong: This paper deepens the understanding of some phenomenon or lowers the barriers to an existing research direction.

**Paper Topic And Main Contributions:**

Edit following author reponse:
------
I thank the authors for their detailed comments, which answer my questions, and I raise my score accordingly. I hope that the clarification here are added to the paper.
------


The paper proposes a method for monolingual token initialization from multilingual language models. This is achieved by training small-scale monolingual LMs, and then expressing tokens which are only in the monolingual model as a linear combination of known embeddings in the multilingual LM. This approach is tested both in intrinsic LM performance, as well as in downstream tasks.

Although I like the approach, and I think it's very clearly described and it's well motivated, I was disappointed by the performance. Unless I missed something, I think that in most of the tested settings, the difference from simpler baselines is well within the error boundary.

While I don't think that performance on its own is the only measure by which to review a paper, I do think that in this case it's (1) much of the paper premise seems predicated on this approach being useful for empirical reasons, and (2) I didn't find the paper acknowledging that this is the case, and trying to provide insights into why this may happen.

If that's indeed the case, I think that the paper would benefit from a deeper look into why the performance is similar, e.g., by examining error patterns via manual evaluation. I think that would have helped me learn more from this paper.

**Questions For The Authors:**

- I didn’t understand Figure 1, what are the different points, what are the dashed lines, what do orange and yellow colors signify? In general, I think it’d be helpful to have a Formal Definitions sections where it’s clearly defined what’s the source model, what’s the target, etc.

- I wonder if comparing loss across different tokenizers is a valid comparison (Table 4), can you please elaborate on how this is done? I think that this correlates with perplexity, which will be biased towards smaller vocabularies as they’re inherently less surprising.

**Reasons To Accept:**

* Interesting and simple approach for a timely problem - how to infer initializations for new tokens when adapting multilingual PLMs to a specific language and domain.

**Reasons To Reject:**

* The performance seems rather similar to the baseline. While this isn't necessarily a problem, it is not discussed and explored in the paper. Conversely, it states that "Without further training, FOCUS significantly outperforms all other initialization methods", but I couldn't find what kind of stat. significance performance tests were performed, as the performance difference in most tables seem to be within the margin of error.

**Reproducibility:**

4: Could mostly reproduce the results, but there may be some variation because of sample variance or minor variations in their interpretation of the protocol or method.

**Reviewer Confidence:**

3: Pretty sure, but there's a chance I missed something. Although I have a good feel for this area in general, I did not carefully check the paper's details, e.g., the math, experimental design, or novelty.

**Typos Grammar Style And Presentation Improvements:**

- I think that the paper is superfluous with footnotes. Consider omitting in a few
- The mathematical font for symbols used in Section 2 are nonstandard I think.

---

> ### Author Rebuttal · Authors · 2023-08-28
>
> We thank you for your thorough review and will address your suggestions on our paper’s presentation.
>
> ---
> Regarding your comments:
>
> We state: “Without further training, FOCUS significantly outperforms all other initialization methods”. This refers to the results reported in Table 4. Here, we can clearly see that FOCUS performs much better than other initialization methods **without further training**.
>
> The results on downstream tasks (Tables 2 & 3) and loss curves (Figure 2) are reported after further training. Indeed, we do see that the gap between FOCUS and other methods is smaller here (also acknowledged in the paper, e.g. l. 376ff). However, we would like to point out the following:
> * Naturally, the effect of initialization is less pronounced, the longer one trains a model. We have deliberately constructed a difficult evaluation with our long training regime of 12.8 billion tokens.
> * For the baseline of random initialization of a new embeddings matrix, we train just the embeddings for an additional 20% longer (which is not done for FOCUS). Nevertheless, we find that FOCUS obtains better results and in most cases also outside the margin of error. Note that random initialization is denoted as 2-Stage-Lapt in Tables 2 & 3 to signify the training regime with an additional 20% training. We will clarify that this uses random initialization.
> * When comparing FOCUS to LAPT of the original model (without vocabulary replacement), it is important to remember that the vocabulary replacement used with FOCUS yields a smaller model with fewer parameters and enables faster training and inference as additional benefits. Also, FOCUS still performs better.
> * In summary: Adapting a multilingual model using the original vocabulary is a strong baseline. However, especially for low-resource languages and crucially for languages that use a different script, customizing the vocabulary for the target language can be helpful. When using vocabulary replacement, we do see that on average, FOCUS improves over random initialization and also over WECHSEL (without requiring a bilingual dictionary).
>
> ---
> Regarding your questions:
>
> Figure 1 is a high-level schematic of FOCUS: For each target token we want to initialize (blue), we select a set of similar other tokens (orange, connected by dashed lines). We colored the tokens that are “not similar” as yellow. Indeed, we sacrificed some clarity for brevity of presentation in Figure 1 and will work to improve this tradeoff.
>
> Yes, comparing losses between different tokenizers would be problematic because in our case perplexity = exp(loss). In Table 4, for each language, all compared methods use the same tokenizer.
>
> ---
> We thank you again for your helpful comments and hope that we have answered your questions sufficiently well for you to consider raising your scores.

---

### Official Review · Reviewer_edfE · 2023-08-04

**Soundness:** 4

**Excitement:**

4: Strong: This paper deepens the understanding of some phenomenon or lowers the barriers to an existing research direction.

**Paper Topic And Main Contributions:**

The paper introduces a method of adapting the embedding layer (and associated vocabulary) of multilingual language models for language-specific downstream tasks.

The method is based on averaging existing learned embeddings of tokens in monolingual vocabulary to obtain better representation for specific languages. The main strength of the method is that it increases the model’s performance on downstream tasks (QA, XNLI, NER)  while parameter count by 55%, thus increasing training/inference speed by 40%.


**Questions For The Authors:**

A. In L167, you mention that overlap stems from code-switching. When applying FOCUS to a language with the unseen script do you always apply code-switching or just rely on code-switched words naturally occurring in the data?

B. What do you mean by “always transferring the Transformer layers of XLM-R”? That in all examined models, encoder layers are copied from the original model?

C. In “Random Initialization,” how is embedding mapping conducted when the sizes of source and target vocabularies are different? Are source embeddings subsampled? Do you consider a case where the source vocabulary would be larger than the target vocabulary?


**Reasons To Accept:**

The method brings overall improvement in downstream tasks and reduces the size of the model leading to faster inference and training on in-language data via the LAPT method.

The introduced method is simple and clearly described. FOCUS requires just an unlabeled corpus of in-language data.

The paper presents an in-depth analysis of the strengths of the introduced method. It is compared with recent strong baselines.


**Reasons To Reject:**

The results in downstream tasks show that FOCUS with LAPT is further than the standard deviation above the original model with LAPT or WECHSEL with LAPT. This strongly suggests that most lifting is done by LAPT, and FOCUS does not offer a significant improvement. Figure 2 shows that good embedding initialization can lead to faster learning on in-language data, yet even for low-resource languages, the difference between FOCUS+LAPT and LAPT is small.

The analysis was performed only on one pre-trained language model (XLM-R). I’m afraid that the improvement stemming from better embedding initialization can be even smaller for bigger-scale models.


**Reproducibility:**

4: Could mostly reproduce the results, but there may be some variation because of sample variance or minor variations in their interpretation of the protocol or method.

**Reviewer Confidence:**

4: Quite sure. I tried to check the important points carefully. It's unlikely, though conceivable, that I missed something that should affect my ratings.

**Typos Grammar Style And Presentation Improvements:**

I find the paper well organized. I have just a few suggestions to move some pieces of information to help the reader better understand the contributions and methodology.

L102: Ad: Reduction of the model size. I suggest underscoring this fact and reiterate it in the experimental setting.

L290 Ad: WECHSEL_EN. I was wondering how the English tokens were identified. The answer came only at the end of the result section (in L492). I suggest bringing this explanation here (maybe in the form of a footnote).

---

> ### Author Rebuttal · Authors · 2023-08-28
>
> We thank you for taking the time to review our paper.
>
> ---
> Regarding your comment about the role of LAPT:
>
> Indeed, further language-adaptive pretraining (LAPT) is important and FOCUS will not work well without it. However, in Tables 2 and 3, we can draw a direct one-to-one comparison of FOCUS initialization vs. random initialization with LAPT (2-Stage-Lapt) for adapting a model with vocabulary replacement. In this comparison, we clearly see that FOCUS obtains better results, in most cases outside the margin of error. When comparing FOCUS (vocabulary replacement) to LAPT with the original vocabulary (line 2 in the Tables), it is important to keep in mind that the model with FOCUS has far fewer parameters due to the replaced vocabulary and is faster and more memory-efficient for training and inference. Still, FOCUS also obtains better results.
>
> ---
> Regarding your questions:
>
> A: We do not add any code-switching but simply take advantage of code-switching if it naturally occurs in the target language data (so if it is very uncommon in a particular target language, we cannot leverage it). However, even without any code-switching and a new, unseen script, FOCUS can also take advantage of named entities, numbers, symbols, and punctuation.
>
> B: Yes, that understanding is correct. We will update the paper to make it more clear.
>
> C: In the experiments reported in the paper, the source vocabulary is always larger than the target vocabulary, in which case we copy the first n embeddings from the source until the target embeddings are filled. If we have a smaller source vocabulary, we can initialize the added embeddings from a normal distribution with mean and standard deviation of the source embeddings, similar to how we deal with target tokens that do not have an auxiliary embedding.
>
> ---
> Furthermore, we will incorporate your suggestions on improving the paper's presentation.
> We hope that our answers are helpful in addressing all of your questions.

---

### Official Review · Reviewer_EJyH · 2023-08-05

**Typos Grammar Style And Presentation Improvements:** fasttext --> fastText
**Soundness:** 4

**Excitement:**

4: Strong: This paper deepens the understanding of some phenomenon or lowers the barriers to an existing research direction.

**Paper Topic And Main Contributions:**

This paper proposes FOCUS, which is an embedding initialization method for the monolingual specialization of language models with a language-specific tokenizer. The authors conduct experiments on NLI, QA and NER tasks and show that this method is able to effectively initialize the embedding matrix for a new tokenizer based on information from the source model's embedding matrix.

**Questions For The Authors:**

1. Could you give more details about how you apply fastText to the unlabeled target language data?
2. What if a word in the target language cannot be found in fastText vocabulary, how do you handle this?
3. Is there any reason you choose fastText rather than other static word embeddings such as GloVe and word2vec?

**Reasons To Accept:**

The paper is well-written and the idea is straightforward. The authors use the vocabulary overlap between source and target languages to effectively transfer the pre-trained embeddings to the new target tokenizer's embedding matrix, and without requiring either bilingual dictionaries or an alignment of embedding spaces for different languages.

**Reasons To Reject:**

From my point of view, I do not have reason to reject it.

**Reproducibility:**

4: Could mostly reproduce the results, but there may be some variation because of sample variance or minor variations in their interpretation of the protocol or method.

**Reviewer Confidence:**

3: Pretty sure, but there's a chance I missed something. Although I have a good feel for this area in general, I did not carefully check the paper's details, e.g., the math, experimental design, or novelty.

---

> ### Author Rebuttal · Authors · 2023-08-28
>
> We thank you for taking the time to review our paper.
>
> Regarding your questions:
>
> 1. To obtain token-level embeddings for a target language, we first tokenize the entire unlabeled language corpus using the language-specific tokenizer. The result is written to disk, with whitespace as a delimiter between individual tokens (although this is an implementation detail of fastText). Then, we apply the standard fastText training procedure. Note that tokenizing first is crucial to obtain token embeddings rather than word embeddings. We provide some further implementation details in Appendix B.
>
> 2. In our experiments, we train language-specific tokenizers on the same corpus that we use for fastText training, so we do not have this problem. However, some tokens might only occur very infrequently, so they will not obtain a reliable fastText embedding. In these cases, we will initialize them from a normal distribution with the mean and standard deviation of the source embeddings.
>
> 3. FOCUS is not at all limited to using fastText token embeddings. However, fastText is more recent than word2vec or GloVE and has also been used by previous work such as Minixhofer et al. 2022. We will also correct the spelling of "fasttext" to "fastText".
>
> We hope that our answers are helpful.

---

### Meta-Review · Area_Chair_myfg · 2023-09-17

**Recommendation:** 4

**Metareview:**

The paper proposes to adapt the embedding layer of multilingual language models for language-specific downstream tasks by averaging existing embeddings of tokens in the target language vocabulary to obtain better initialization. The approach seems to lead to pretty significant improvements while reducing parameter count.

---

### Decision · Program_Chairs · 2023-10-07

**Decision:**

Accept-Main

**Comment:**

The paper proposes to adapt the embedding layer of multilingual language models for language-specific downstream tasks by averaging existing embeddings of tokens in the target language vocabulary to obtain better initialization. The approach seems to lead to pretty significant improvements while reducing parameter count.